# Identification of Two Novel Circular RNAs Deriving from *BCL2L12* and Investigation of Their Potential Value as a Molecular Signature in Colorectal Cancer

**DOI:** 10.3390/ijms21228867

**Published:** 2020-11-23

**Authors:** Paraskevi Karousi, Pinelopi I. Artemaki, Christina D. Sotiropoulou, Spyridon Christodoulou, Andreas Scorilas, Christos K. Kontos

**Affiliations:** 1Department of Biochemistry and Molecular Biology, Faculty of Biology, National and Kapodistrian University of Athens, 15701 Athens, Greece; pkarousi@biol.uoa.gr (P.K.); partemaki@biol.uoa.gr (P.I.A.); xristswt@biol.uoa.gr (C.D.S.); ascorilas@biol.uoa.gr (A.S.); 2Fourth Surgery Department, University General Hospital “Attikon”, 12462 Athens, Greece; spyridon.christodoulou@yahoo.gr

**Keywords:** circRNAs, alternative splicing, transcriptomics, miRNA sponges, RNA-binding proteins, apoptosis, colon adenocarcinoma, tumor biomarker, CRC prognosis, TNM stage

## Abstract

The utility of circular RNAs (circRNAs) as molecular biomarkers has recently emerged. However, only a handful of them have already been studied in colorectal cancer (CRC). The purpose of this study was to identify new circRNAs deriving from *BCL2L12*, a member of the BCL2 apoptosis-related family, and investigate their potential as biomarkers in CRC. Total RNA extracts from CRC cell lines and tissue samples were reversely transcribed. By combining PCR with divergent primers and nested PCR followed by Sanger sequencing, we were able to discover two *BCL2L12* circRNAs. Subsequently, bioinformatical tools were used to predict the interactions of these circRNAs with microRNAs (miRNAs) and RNA-binding proteins (RBPs). Following a PCR-based pre-amplification, real-time qPCR was carried out for the quantification of each circRNA in CRC samples and cell lines. Biostatistical analysis was used to assess their potential prognostic value in CRC. Both novel *BCL2L12* circRNAs likely interact with particular miRNAs and RBPs. Interestingly, circ-BCL2L12-2 expression is inversely associated with TNM stage, while circ-BCL2L12-1 overexpression is associated with shorter overall survival in CRC, particularly among TNM stage II patients. Overall, we identified two novel *BCL2L12* circRNAs, one of which can further stratify TNM stage II patients into two subgroups with substantially distinct prognosis.

## 1. Introduction

Colorectal cancer (CRC) is a malignancy developed in the colon or rectum. Of all annually diagnosed cancers and cancer-related deaths, CRC represents approximately 10%, as it is the second most common cancer diagnosed in women and the third most common one in men [1]. Due to the high mortality rate of CRC, several attempts have been made towards the elucidation of its molecular background. Three specific gene categories are considered to play a basic role in CRC development: tumor suppressor genes such as APC, DCC, TP53, and the SMAD protein family; oncogenes such as KRAS and NRAS; and DNA repair genes [2]. Moreover, CRC cells are characterized by great molecular heterogeneity: their development is followed by loss of genomic integrity, accumulation of mutations, chromosomal and microsatellite instability, methylation of the DNA, and DNA repair defects. Moreover, signaling pathways that are vital for the cells, such as WNT/β-catenin, PI3K/AKT, and TGF-β/SMAD, are deregulated and thus, contribute to CRC progression [3,4]. The complex molecular background is a large impediment in CRC understanding.

Apoptotic-like programmed cell death, or apoptosis, is a tightly regulated process, involved in many cellular processes, including morphogenesis, aging, and tissue homeostasis [5,6]. Disruption of the apoptotic mechanism may lead to tumor formation. Apoptosis is mediated via two distinct pathways: the intrinsic or mitochondrial pathway, triggered by intracellular signals, and the extrinsic or death receptor pathway, triggered by extracellular signals [7]. The strict regulation of the intrinsic apoptotic pathway is partly achieved through the action of the BCL2 family proteins [8]. BCL2 family members can be characterized as pro- or anti-apoptotic, depending on their effect on the pathway. These proteins contain one or more of the four conserved homology domains, namely BCL2 homology (BH) domains. BH1 or BH2 domains can be found on anti-apoptotic proteins and pro-apoptotic proteins, as well. BH3 domains represent a characteristic of pro-apoptotic proteins, while BH4 domains can be found on anti-apoptotic family members [9].

*BCL2L12* is a member of the BCL2 family, which was identified and described by members of our research group [10]. The *BCL2L12* gene covers an 8.8 kb genomic area on chromosome 19q13.3. Its pre-mRNA structure is composed of seven exons and six introns; a great number of splice variants have been identified by members of our research group [11,12]. The protein isoform encoded by the main transcript is a proline-rich protein that can interact with tyrosine kinases and thus, exhibit oncogenic capacity [13,14]. The main BCL2L12 protein isoform bears a BH2 and a BH3-like domain; although it does not bear a typical anti-apoptotic protein structure [15], it has been reported to exert anti-apoptotic activity through the inhibition of caspases 3 and 7 and p53 binding [16]. Members of our research group have also assessed the role of *BCL2L12* mRNA in various types of solid tumors and hematological malignancies [17,18,19,20,21,22], including CRC. *BCL2L12* mRNA was reported to be abundant in CRC and was characterized as a molecular indicator of favorable prognosis for CRC patients [12,23,24].

Circular RNAs (circRNAs) constitute an RNA type obtained via a head-to-tail splicing event called back-splicing. Although they were discovered in RNA viroids in the 1970s, circRNAs have been neglected so far, as they were considered as byproducts of splicing events [25]. Recent advances in high-throughput sequencing and circRNA-specific bioinformatic tools revealed the widespread expression of circRNAs in eukaryotic cells, arousing scientific interest. At first, circRNAs were characterized as non-coding and their function was thought to be only regulatory [26]; however, some of them have been reported to encode peptides [27]. They have been found to act as microRNA (miRNA) sponges and subsequently, protect mRNAs from miRNA-dependent degradation and translation repression [28]. Interactions of circRNAs with proteins have been detected as well; some of them contain RNA-binding protein (RBP) motifs and thus, act as protein sponges, decoys, or recruiters [29]. Most circRNAs derive from known protein-coding genes and consist of either a single or multiple exons (EcircRNAs), both exons and retained introns (EIciRNAs), or even only introns (ciRNAs) [30]. However, our knowledge concerning circRNAs transcribed from apoptosis-related genes is poor.

This study aimed to further investigate the role of *BCL2L12* in the molecular background of CRC, by identifying circRNAs deriving from this gene and assessing their role as molecular biomarkers. The implication of circRNAs in CRC [2,31], the proven overexpression of *BCL2L12* in CRC, and its implication in apoptosis and tumorigenesis advocated for the investigation of *BCL2L12* circRNAs in this context [24]. Through this study, two novel circRNAs deriving from *BCL2L12* were identified; their role as molecular biomarkers in CRC was assessed, while bioinformatical tools were used, in order to predict their interactions.

## 2. Results

### 2.1. Identification of Two Novel circRNAs Deriving from BCL2L12

Two novel circRNAs deriving from *BCL2L12* were identified. Both of them comprise only exons. The first one, which will be termed as circ-BCL2L12-1, comprises truncated exons 6 and 1, whole exons 2, 3, and 4, and the extended exon 5, which was previously identified by members of our research group [11]; the length of this circRNA is 501 nucleotides (nt) and the back-splice event takes place between exons 6 and 1. The second one, which will be termed as circ-BCL2L12-2, consists of truncated exon 6 and whole exons 2, 4, and 5. It is important to note that the second circRNA contains a longer part of exon 6, compared to the first one. The length of this circRNA is 442 nt and the back-splice event takes place between exons 6 and 2. The exon structures and sequences of both circRNAs are shown in Figure 1.

### 2.2. Putative Interactions of BCL2L12 circRNAs with miRNAs and RBPs

In silico analysis showed that circ-BCL2L12-1 is predicted to sponge seven miRNAs, while circ-BCL2L12-2 is predicted to sponge five miRNAs. Out of these 12 miRNAs, only miR-1915-5p had a high prediction score for binding to circ-BCL2L12-1. These miRNAs, the prediction scores, and the binding motifs of the circRNAs are shown in Table 1. The targets of miR-1915-5p, which were found in at least two of the four tools and have been reported to play a role in CRC, are shown in Appendix A.

In both circRNAs, various protein-binding sites were detected and many RBPs were predicted to bind to them. RBPmap provided more information about the number of RBPs and their binding sites compared to beRBP; thus, we chose to use the results observed by this tool. We selected those RBPs with more than five binding sites, accompanied by a high probability score; these are presented in Table 2.

circ-BCL2L12-2 is also predicted to have an open reading frame (ORF) and be subjected to translation. This ORF is marked in Figure 1f. However, no internal ribosomal entry sites (IRES) were predicted.

### 2.3. Standardization of Real-Time qPCR Assays for BCL2L12 circRNA Quantification

A real-time qPCR assay was standardized for the specific *BCL2L12* circRNA quantification. Aiming to assure the quantitative results of each assay, a standard curve was generated for each amplicon, using serial dilutions of PCR products, deriving from a 25-cycle PCR assay. For circ-BCL2L12-1 and circ-BCL2L12-2 standard curves, PCR products from pre-amplification of Caco-2 cell line cDNA were serially diluted. For the *HPRT1* standard curve, cDNA from a fresh frozen tissue specimen was pre-amplified and serially diluted. The efficiencies of the standard curves for *HPRT1*, circ-BCL2L12-1, and circ-BCL2L12-2 were 91%, 96%, and 107%, respectively, and therefore, the prerequisites for the quantification of our results were fulfilled. The standard curves of circ-BCL2L12-1 and circ-BCL2L12-2 are shown in Figure 2a,b, respectively. Moreover, each melt curve was unique, indicating the specificity of the amplicon. The melt curves of circ-BCL2L12-1 and circ-BCL2L12-2 are shown in Figure 2c,d, respectively.

### 2.4. Expression Analysis of BCL2L12 circRNAs in CRC Cell Lines

Through the pre-amplification and qPCR assays, we relatively quantified the expression levels of both circRNAs in Caco-2, HCT 116, HT-29, COLO 205, SW 620, DLD-1, and RKO CRC cell lines. circ-BCL2L12-1 is present in all CRC cell lines, while circ-BCL2L12-2 was detected in two of them, namely Caco-2 and HCT 116. The expression levels of circ-BCL2L12-1 in these two cell lines were higher than those of circ-BCL2L12-2. The amplification plots of the qPCR assays for both circRNAs are shown in Figure 3.

### 2.5. Expression of BCL2L12 circRNAs in Malignant Tumors and Non-Cancerous Tissues

The distributions of *BCL2L12* circRNAs did not differ significantly between malignant tumors and paired non-cancerous tissues (Appendix A). The mean expression values of both *BCL2L12* circRNAs and mean values of other scale variables are shown in Table 3.

### 2.6. Association of Circ-BCL2L12-2 Expression with TNM II and III Stages

The CRC patients were stratified into two distinct subgroups (high vs. low), according to the expression levels of each circRNA, determined in relative quantification units (RQUs). The median expression of each circRNA was used as a cut-off point. The frequencies of circ-BCL2L12-2–high and –low patients varied significantly between TNM II and TNM III stages. Specifically, the frequencies of circ-BCL2L12-2–low patients were significantly higher in the TNM III stage compared to the TNM II stage, in contrast to circ-BCL2L12-2–high patients (*p =* 0.007) (Appendix A).

### 2.7. Circ-BCL2L12-1 as a Potential Molecular Indicator of Poor Prognosis in CRC

Survival data were available for 151 out of the 168 CRC patients. Kaplan–Meier survival analysis revealed the prognostic potential of circ-BCL2L12-1, concerning overall survival (OS); specifically, CRC patients with higher levels of circ-BCL2L12-1 showed significantly shorter OS intervals, compared to those with lower levels (*p* = 0.036; Figure 4a). Additionally, similar results regarding OS were shown in the subgroup of TNM stage II patients (*p* = 0.045; Figure 4b). These results were strengthened by univariate Cox regression analysis, which showed a hazard ratio (HR) of 1.92 for patients with high circ-BCL2L12-1 levels [bias-corrected and accelerated (BCa) 95% confidence interval (CI) = 1.01–3.76, bootstrap *p* = 0.035; Table 4].

## 3. Discussion

By virtue of the advances in RNA sequencing (RNA-seq) and other new technological approaches, circRNAs have gained ground in molecular and cellular research, as they exert various biological functions. Some of these functions, such as transcription regulation, protein binding, and miRNA-sponging, can be crucial for cancer development. Various studies have implied their role in cancer [32], and particularly, in CRC, a deregulated expression pattern of circRNAs has been reported, compared to normal tissues [33].

Prompted by our interest in circRNAs, we first attempted to seek, in the circBase, RNA-seq data that prove the existence of circRNAs deriving from *BCL2L12.* Indeed, seven entries concerning *BCL2L12* circRNAs were available [34]; however, none of these circRNAs have been experimentally validated yet, while the circRNAs identified in the current study have not been submitted to any of the available databases. Thus, no prediction data were available for the circRNAs identified in this study.

Many studies have declared that circRNAs are formed by exons that are abundant in the mRNA transcripts derived by the same gene [35]. However, the present study showed that both novel EcircRNAs include truncated exons, which have not been detected in the mRNAs transcribed from *BCL2L12,* despite the great number of existing splice variants [11,12]. Interestingly the truncated exons of both circRNAs are located in the back-splice site (exons 6 and 1 in circ-BCL2L12-1; exon 6 in circ-BCL2L12-2). Moreover, the back-splice site in both cases is not a canonical one; this could be attributed either to the cancer state of the cells [36], or to a circRNA biogenesis mechanism, which has not been elucidated yet.

Attempting to predict the interactions of the aforementioned circRNAs with other molecules and understand their function, we concluded that both of them are predicted to bind to splicing factors. This is expectable, due to the implication of specific splicing factors, including MNBL1, in circRNA biogenesis [37]. Interestingly, a linkage between circRNAs and splicing factors has previously emerged, affecting cancer development. More specifically, circSMARCA5 has been reported to inhibit the migration of glioblastoma multiforme cells through a molecular axis including SRFS1/SFFS3/PTB [38], indicating that the binding of splicing factors may be involved in other functions besides biogenesis. Moreover, a deregulated expression pattern of splice factors has been reported in CRC and in cancer generally, for instance, PTB1 and CELF1 have been reported to play a role in CRC proliferation [39,40]. Since all the aforementioned factors are predicted to bind to these novel *BCL2L12* circRNAs, it is important that these interactions are experimentally verified and further elucidated.

As far as miRNAs are concerned, miR-1915-5p, which is predicted to be sponged by circ-BCL2L12-1 as shown in Table 1, is also predicted to target *IGF2BP1*, *PRDX3, HDAC2, DAPK1,* and *EAF1* (Appendix A). Particularly, HDAC2 was reported to facilitate tumorigenesis in CRC via chromatin structure regulation, while PRDX3 is overexpressed in CRC stem cells, and is involved in tumorigenesis as well. It seems to exert its function by maintaining the proper mitochondrial function in CRC stem cells and subsequently, promotes their survival [41,42]. On the other hand, DAPK1 acts as a tumor suppressor, through the inhibition of TACSDT2, a receptor that transduces Ca2+ signals and subsequently, leads to enhanced proliferation, invasion, and self-renewal signals [43]. EAF1 serves as a tumor suppressor as well, via the regulation of the canonical WNT/β-catenin pathway, which plays a key role in CRC development [44,45]. A tumor-suppressive function is exerted by IGF2BP1 too, which inhibits KRAS, CDC34, and MYC expression and promotes apoptosis [46]. As miR-1915-5p is predicted to be sponged by circ-BCL2L12-1, all the aforementioned targets are likely to be upregulated, due to the lack of miRNA-mRNA binding and the subsequent inhibition of the mRNA expression. This information suggests a multifarious impact of circ-BCL2L12-1 on CRC cells; however, further investigation is required.

In the current study, the prognostic potential of these circRNAs in CRC was also assessed. Due to their increased stability, which is attributed to their circular structure, circRNAs could serve as great molecular biomarkers. Although circ-BCL2L12-2 was not proved to have prognostic value and was not detected in many CRC samples and most cell lines, it was inversely associated with the TNM stage; thus, its biological function merits further examination. On the contrary, circ-BCL2L12-1 was more abundant and was also shown to predict the OS of the patients in our cohort (Figure 4a).

Interestingly, circ-BCL2L12-1 was shown to predict the OS of TNM II patients. The survival intervals of TNM II stage CRC patients often differ significantly. Therefore, studies have focused on the molecular mechanisms underlying CRC, in order to establish a stratification system that describes the clinical stage and predicts the outcome of these patients more efficiently. One step towards the establishment of a new stratification system was made with the formation of consensus molecular subtyping (CMS). CMS is a stratification system based on the biological characteristics of CRC patients that has recently emerged [47]. Recently, Purcell et al. suggested that consensus molecular subtyping is used, as it improves the prognosis of TNM II stage CRC patients [48]. Our results show that TNM II stage patients can be sub-grouped into two distinct strata, with different survival probability: patients with lower circ-BCL2L12-1 levels show longer OS intervals, close enough to those of TNM I stage patients, while patients with higher circ-BCL2L12-1 levels show shorter OS intervals, even shorter than those of TNM III stage patients (Figure 4b). Thus, circ-BCL2L12-1 could be included in a molecular stratification of CRC patients.

Summarizing the key findings of this study, the existence of circRNAs deriving from BCL2L12 was revealed; both circRNAs consist of known exons and truncated ones. circ-BCL2L12-2 is predicted to have one functional ORF, but no IRES are predicted. Additionally, both circRNAs are predicted to bind miRNAs and RBPs, as revealed by the miRDB and RBPmap tools, and shown in Table 1 and Table 2, respectively. The deciphering of their interactions can provide new insights and be a step towards the understanding of the molecular background of CRC. The identification of peptides translated by these molecules is also a putative study field, as it would be interesting to examine functional similarities between those peptides and proteins encoded by linear transcripts. The biostatistical analysis conducted revealed that circ-BCL2L12-1 could be used as a molecular biomarker of poor prognosis for the OS of CRC patients and can provide a better stratification for TNM II patients based on their OS intervals. Moreover, it would be interesting to assess putative associations of these novel circRNAs and other molecular characteristics of the patients, for instance, *RAS* or *BRAF* or *TP53* mutations, and MMR deficiency. Additionally, the existence of these molecules can be investigated in other tissues, to examine if their expression pattern is tissue-specific or not. 

Our future goals include the elucidation of the biological functions of the circRNAs identified and the investigation of other circRNAs deriving from apoptosis-related genes. Moreover, their dynamic as molecular biomarkers has not been fully elucidated yet; as circRNAs have been detected in exosomes, it would be interesting to investigate the abundancy of *BCL2L12* circRNAs in these vesicles and evaluate their screening potency. As exosomes are easily accessible in biological fluids, screening molecular biomarkers could gradually replace the highly invasive colonoscopy.

## 4. Materials and Methods

### 4.1. Cell Culture

Seven CRC cell lines, namely Caco-2, HCT 116, HT-29, COLO 205, SW 620, DLD-1, and RKO, were cultivated. The mutational status of the CRC cell lines is shown in Appendix A [49]. All cells were seeded at a concentration of 1 × 10^5^ cells/mL and incubated in a humidified atmosphere for 24 h. The CO_2_ concentration was adjusted to 5% and the temperature to 37 °C. The appropriate medium was chosen for each cell line, following the guidelines of ATCC^®^.

### 4.2. Tissue Sample Collection

One hundred and sixty-eight malignant colorectal tumors and 63 paired non-cancerous colonic fresh frozen tissue specimens were provided by the University General Hospital “Attikon”. The patients were followed up for 52 months (median time) and information regarding disease-free (DFS) and OS was collected, for 151 out of the 168 patients. The median age of the CRC patients was 69 years (range: 35–93). The clinicopathological characteristics of the patients are described in Table 5. This research study was conducted in compliance with the 1964 Declaration of Helsinki and its later amendments and was approved by the institutional Ethics Committee of the University General Hospital “Attikon”, Athens, Greece (approval number: 31; 29 January 2009). Moreover, written informed consent was obtained from all participants.

### 4.3. RNA Extraction and Reverse Transcription

Total RNA extraction from CRC cell lines and samples was carried out using the TRIzol^®^ Reagent (Ambion™, Thermo Fisher Scientific Inc., Waltham, MA, USA). The concentration and purity of each RNA sample were evaluated spectrophotometrically. Subsequently, 2 µg of each RNA sample was subjected to reverse transcription using M-MLV reverse transcriptase (Invitrogen™, Thermo Fisher Scientific Inc.) and random hexamers (New England Biolabs Ltd., Hitchin, UK), following the manufacturers’ instructions.

### 4.4. Primer Designing and PCR

The CRC cell lines were mixed in equal volumes to create a CRC pool. This pool was generated for experimental purposes, as this way, we were able to identify circRNAs expressed in any of the CRC cell lines. Divergent primers were designated for *BCL2L12*, so that only circular and not linear RNAs would be amplified during the PCR assay (Figure 1a,b) [50]. The CRC pool was subjected to a first-round PCR assay, which was conducted using KAPA Taq DNA Polymerase (KAPA Biosystems Inc., Woburn, MA, USA) in a MiniAmp Thermal Cycler (Applied Biosystems™, Thermo Fisher Scientific Inc.). The reaction mix contained 19.4 µL nuclease-free H_2_O, 2.5 µL 10x KAPA Taq Buffer, 0.5 µL of a dNTP mix (containing each dNTP at an initial concentration 10 mM), 1 µL of each primer (initial concentration 10 µM), 0.1 µL KAPA Taq DNA Polymerase (initial concentration 5 U/µL), and 0.5 µL of cDNA pool template. The thermal protocol was conducted as follows: An initial denaturation step at 95 °C for 3 min, followed by a cycling step, carried out for 35 cycles, consisting of a denaturation step at 95 °C for 30 s, an annealing step at 60 °C for 30 s, and an elongation step at 72 °C for 30 s. A final elongation step was carried out at 72 °C for 1 min.

Due to the low expression levels of these circRNAs compared to the linear RNAs, second-round (semi-nested) PCR assays were conducted, after diluting the PCR products at a ratio of 1:50 in nuclease-free H_2_O (Figure 1c,d) [51]. The semi-nested PCR assays were conducted as described above. The sequences of the primers used for both assays are listed in Appendix A, while the primer pairs and sizes of the amplicons observed are listed in Appendix A.

### 4.5. Agarose Gel Electrophoresis and Sanger Sequencing

After the PCR assays, the PCR products observed through the amplification of the cDNA pool were loaded on a 2% agarose gel, stained with ethidium bromide. After visualizing in UV light, the desired bands were excised, and DNA extraction was carried out using spin columns (MACHEREY-NAGEL GmbH & Co. KG, Düren, Germany). The concentration of the extracted PCR product was evaluated using a Qubit fluorometer (Thermo Fisher Scientific, Inc., Waltham, MA, USA). Subsequently, the sequence was identified by Sanger sequencing (Appendix A, Data Set 1).

### 4.6. Bioinformatical Analysis for Prediction of circRNA Interactions with miRNAs and RBPs

In order to elucidate the interactions between circRNAs and miRNAs, miRDB was used [52]. This tool allows the user to submit a custom RNA sequence and provides a list of miRNAs that bind to this sequence, as well as a probability score. After choosing those miRNAs showing a high probability score (≥70) we searched for other putative targets, using miRDB, miRWalk, TargetRank, and TarBase (v.8) [53,54,55]. We then selected the targets that were identified in at least 2 of the tools and have been reported to play a role in CRC, as shown in Appendix A.

Additionally, 2 more tools were used to investigate the interactions of these circRNAs with RBPs: RBPmap and beRBP [56,57]. Both tools use a custom RNA sequence as an input and provide a list of RBPs, the binding sites, and a probability score. Subsequently, we search for ORFs in the circRNAs to investigate if they could be translated into peptides. Due to the lack of 5’cap in circRNAs, we also searched for the existence of IRES. ORFs were investigated using ORFfinder, while IRESpy was utilized to search for IRES [58].

In order to carry out these predictions, the sequence of the circRNA should be submitted as linear. Therefore, we should “cut” the sequence in a stochastic position. Aiming to assure that no information for binding sites will be lost, due to this decision, we conducted all of the predictions twice, following the cleavage of circRNA sequence at a different point. More specifically, the last exon of the sequence which was submitted the first time in each prediction was transferred at the beginning of the sequence for the following predictions.

### 4.7. Pre-Amplification and qPCR

After the identification of the circRNAs, we aimed to quantify them in cell lines and human tissue sample cDNAs. Due to the low expression levels of the circRNAs, a PCR assay was conducted for the pre-amplification of these molecules and of *HPRT1* mRNA, which was used as a reference for the qPCR assay. After the execution of the PCR assay in 15, 20, 25, 30, and 35 cycles, 25 cycles were chosen as the optimal pre-amplification condition. The thermal protocol was carried out as described above. Prior to the qPCR assay, the PCR products were diluted at a ratio of 1:50 in nuclease-free H_2_O.

A real-time qPCR assay was developed. In brief, the reaction, which contained KAPA SYBR FAST qPCR Master Mix (2X) Kit (KAPA Biosystems Inc.), was performed in an ABI 7500 Fast Real-Time PCR System (Applied Biosystems™), following a standard thermal protocol for cycling and melting, as previously described [59,60]. A standard curve was generated for each amplicon, using serial dilutions of PCR products. A graph was built by plotting the threshold cycle (C_T_) versus the quantity. Each qPCR reaction was performed in duplicates, to assure the reproducibility of the obtained data. The expression levels of each circRNA were calculated using the comparative C_T_ (2^−∆∆CT^) method [61,62]. The relative circRNA expression of each sample was determined in RQUs by calculating the ratio of each circRNA to *HPRT1* molecules divided by the same ratio calculated for the calibrator (Caco-2 cells). The sequences of the primers used are listed in Appendix A, while the primer pairs and sizes of the amplicons observed are listed in Appendix A.

### 4.8. Biostatistical Analysis

Due to the fact that the distribution in our cohort was not normal, non-parametric tests such as the Wilcoxon signed-rank and Mann–Whitney *U* tests were used. Both of them do not take into consideration the distribution in the cohort. After performing descriptive statistics, the Wilcoxon signed-rank test was used to evaluate the difference in the expression of the circRNAs between cancer and non-cancerous samples. The CRC patients were then designated as “high” or “low”, depending on the expression of each circRNA, calculated in RQUs; the separation of the two groups was based on the median value of each circRNA expression. Associations between the expression of each circRNA and other categorical variables were evaluated, using a two-tailed chi-square test.

Subsequently, survival analysis was conducted; Kaplan–Meier curves, concerning DFS and OS, were built, and differences between them were assessed with the log-rank (Mantel–Cox) test. This analysis was carried out in distinct patient groups as well, stratified based on their clinicopathological features. Last, bootstrap univariate and multivariate Cox regression analyses were carried out.

## Figures and Tables

**Figure 1 ijms-21-08867-f001:**
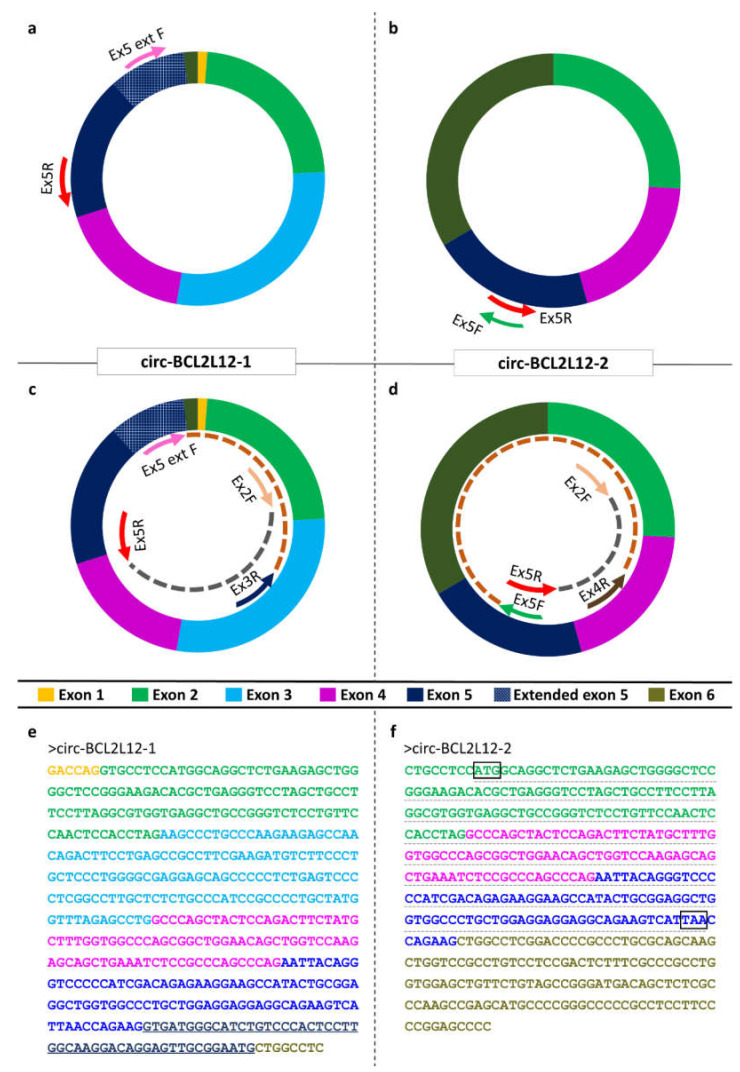
The structure and sequences of *BCL2L12* circular RNAs (circRNAs). (**a**,**b**) Arrows outside the circles show the annealing positions of the primers used in the first-round PCR assay. (**c**,**d**) Arrows inside the circles indicate the annealing positions of the primers used in the semi-nested PCR assays, while arcs indicate the respective amplicons. (**e**,**f**) The sequences of *BCL2L12* circRNAs, starting from the back-splice site. The colored fonts of the exon sequences match the colors used in the graphical illustrations of the circRNA exon structures. The black boxes show the putative translation start and stop codons of the predicted open reading frame (ORF). The ORF is underlined with a grey dashed line.

**Figure 2 ijms-21-08867-f002:**
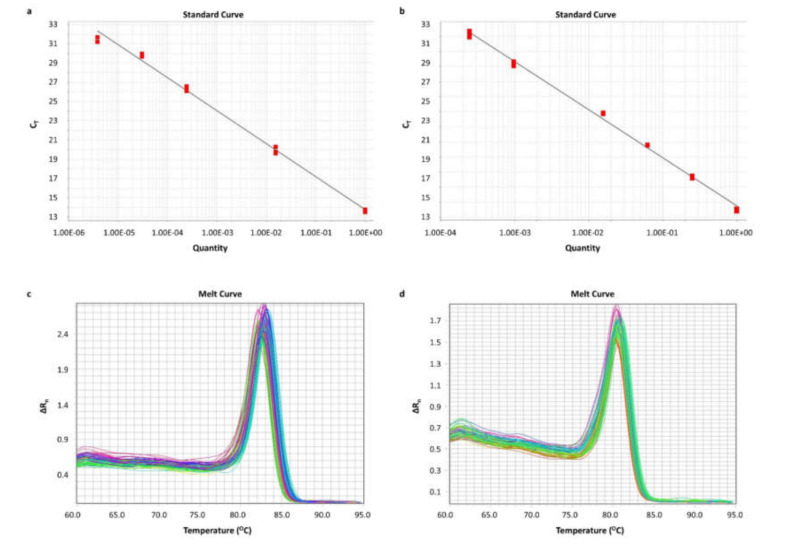
Standard curves built by plotting the threshold cycle (C_T_) versus the dilution, and melt curves observed. For the standard curves, PCR products generated from pre-amplification of Caco-2 cDNA were used. (**a**) Standard curve for circ-BCL2L12-1; 8-fold serial dilutions of the Caco-2 PCR product were used. (**b**) Standard curve for circ-BCL2L12-2; 4-fold serial dilutions of the Caco-2 PCR product were used. (**c**) Melt curve of circ-BCL2L12-1; (**d**) Melt curve of circ-BCL2L12-2.

**Figure 3 ijms-21-08867-f003:**
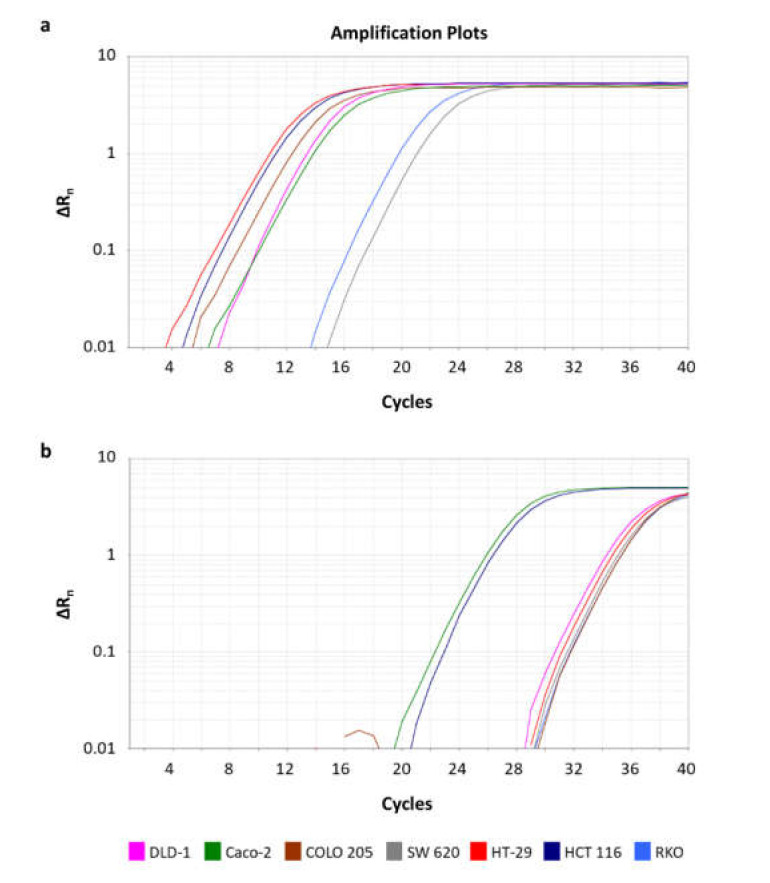
Amplification plots of circ-BCL2L12-1 (**a**) and circ-BCL2L12-2 (**b**) in colorectal cancer (CRC) cell lines.

**Figure 4 ijms-21-08867-f004:**
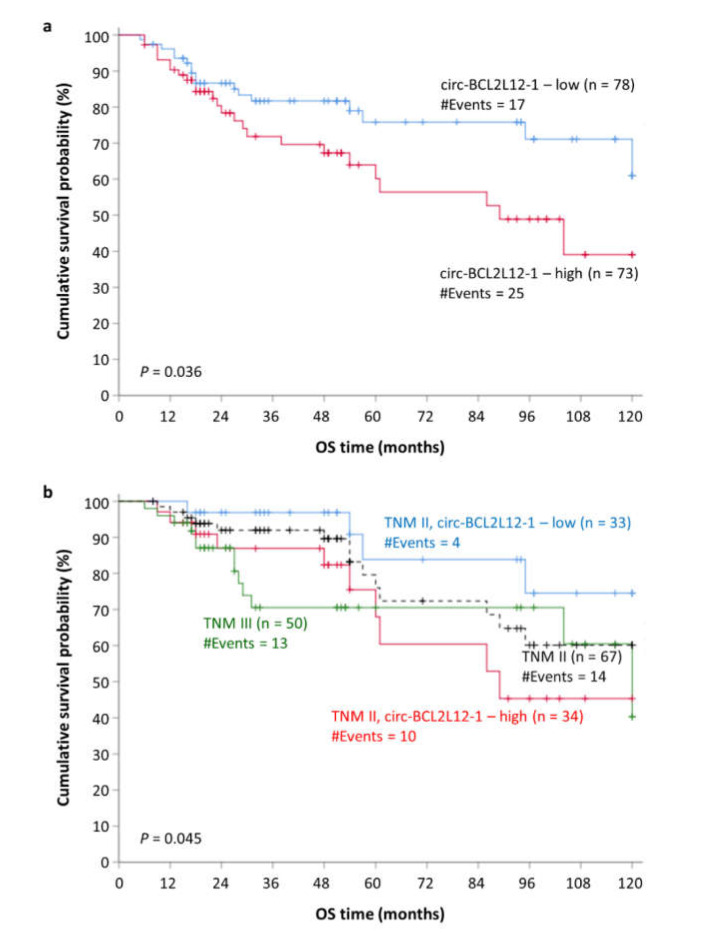
Kaplan–Meier overall survival (OS) curves for CRC patients. (**a**) All CRC patients, categorized according to circ-BCL2L12-1 expression; circ-BCL2L12-1–high patients showed significantly poorer OS compared to the circ-BCL2L12-1–low ones. (**b**) TNM stage II patients, categorized according to circ-BCL2L12-1 expression; circ-BCL2L12-1–high patients showed significantly poorer OS compared to the circ-BCL2L12-1–low ones (the *p*-value refers to this comparison).

**Table 1 ijms-21-08867-t001:** MicroRNAs (miRNAs) that are predicted to be sponged by *BCL2L12* circular RNAs (circRNAs).

*BCL2L12* circRNA	miRNAs Binding to circRNA	Prediction Score ^1^	Binding Motifs
circ-BCL2L12-1	miR-1915-5p	86	GGCAAGGA
	miR-6721-5p	62	CCTGCCCA
	miR-6822-3p	59	CCCTGCTA/CCCTGCT
	miR-6510-5p	57	GAAGCCA/AGAAGCC
	miR-1237-3p	54	AGAAGGA/CAGAAGG
	miR-6815-3p	52	AGAGCCA/TAGAGCC
	miR-6849-3p	50	GGCTGGA/AGGCTGG
circ-BCL2L12-2	miR-4767	59	CGCCCG
	miR-4763-5p	51	GGCAGG
	miR-6729-3p	51	CTCGCCCA
	miR-4649-5p	51	CTCGCCCA
	miR-6849-3p	50	GGCTGGA/AGGCTGG

^1^ miRDB was used for the prediction of miRNAs binding to each *BCL2L12* circRNA.

**Table 2 ijms-21-08867-t002:** RNA-binding proteins (RBPs) that are predicted ^1^ to bind to each *BCL2L12* circRNA.

RBP	circ-BCL2L12-1			circ-BCL2L12-2		
	Number of Binding Sites	Z-Score ^2^	*p*-Value ^3^	Number of Binding Sites	Z-Score ^2^	*p*-Value ^3^
CUGBP Elav-like family member 1 (CELF1)	24	2.54	*0.006*	22	3.70	*0.001*
FUS RNA-binding protein (FUS)	0	–	–	5	3.60	<*0.001*
Muscleblind-like splicing regulator 1 (MBNL1)	31	2.40	*0.008*	25	2.40	*0.008*
Sterile alpha motif domain-containing 4A (SAMD4A)	6	3.56	<*0.001*	6	3.56	<*0.001*
Serine and arginine-rich splicing factor 1 (SRSF1)	21	3.69	<*0.001*	22	3.81	<*0.001*
Serine and arginine-rich splicing factor 2 (SRSF2)	27	3.31	<*0.001*	21	3.31	<*0.001*
Serine and arginine-rich splicing factor 3 (SRSF3)	54	2.92	*0.001*	55	3.29	<*0.001*
Serine and arginine-rich splicing factor 5 (SRSF5)	9	2.61	*0.004*	6	2.22	*0.013*
Polypyrimidine tract binding protein 1 (PTBP1)	20	3.23	<*0.001*	11	2.58	*0.005*
Heterogeneous nuclear ribonucleoprotein H1 (HNRNPH1)	6	3.70	*0.003*	6	2.74	*0.003*

^1^ RBPmap was used for the prediction of RBPs binding to each *BCL2L12* circRNA. ^2^ Defines a significant match. For each RBP, only the highest Z-score is shown. ^3^ Probability of obtaining a specific Z-score. Statistically significant *p*-values are shown in italics.

**Table 3 ijms-21-08867-t003:** Distributions of *BCL2L12* circRNA expression levels in cancerous and non-cancerous tissue samples.

Variable.	Mean ± SE ^1^	Range	Percentiles		
			25th	50th (Median)	75th
**circ-BCL2L12-1 expression (RQU ^2^)**					
in malignant tumors (*n* = 168)	6.03 ± 2.05	0.001–246.8	0.096	0.52	2.25
in non-cancerous tissues (n = 63)	8.46 ± 3.39	0.001–167.3	0.10	0.48	2.93
**circ-BCL2L12-2 expression (RQU ^2^)**					
in malignant tumors (*n* = 60)	2.16 ± 0.76	0.001–40.82	0.035	0.52	1.51
in non-cancerous tissues (*n* = 27)	1.30 ± 0.44	0.004–9.49	0.034	0.15	1.16

^1^ Standard error. ^2^ Relative quantification unit.

**Table 4 ijms-21-08867-t004:** Expression status of *BCL2L12* circRNAs and overall survival (OS) of colorectal cancer (CRC) patients.

Covariate	Univariate Analysis (*n* = 168)			Multivariate Analysis ^1^ (*n* = 168)		
	HR ^2^	BCa 95% Bootstrap CI ^3^	Bootstrap *p*-Value ^4^	HR ^2^	BCa 95% Bootstrap CI ^3^	Bootstrap *p*-Value ^4^
**circ-BCL2L12-1 expression status**						
Low (*n* = 88)	1.00			1.00		
High (*n* = 80)	1.92	1.01–3.76	*0.035*	1.74	0.80–3.90	0.14
**circ-BCL2L12-2 expression status**						
Low (*n* = 108)	1.00					
High (*n* = 60)	0.81	0.38–1.55	0.56			
**Tumor site**						
Colon (*n* = 111)	1.00			1.00		
Rectum (*n* = 57)	1.99	0.98–4.05	*0.028*	1.63	0.71–3.53	0.16
**Tumor sidedness**						
Left (*n* = 115)	1.00					
Right (*n* = 53)	0.79	0.36–1.46	0.49			
**Histological grade**						
I (*n* = 15)	1.00			1.00		
II (*n* = 129)	0.61	0.17–3.67	0.29	0.44	0.12–1.61	0.11
III (*n* = 24)	1.60	0.39–12.96	0.40	1.03	0.22–5.66	0.97
**Venous invasion**						
Absent (*n* = 118)	1.00					
Present (*n* = 19)	1.58	0.43–3.80	0.35			
**Lymphatic invasion**						
Absent (*n* = 121)	1.00					
Present (*n* = 16)	1.61	0.65–3.20	0.25			
**TNM stage**						
I (*n* = 20)	1.00			1.00		
II (*n* = 72)	1.33	0.34–4.0 × 10^4^	0.62	1.69	0.39–4.7 × 10^4^	0.37
III (*n* = 59)	2.02	0.53–5.1 × 10^4^	0.25	2.79	0.70–7.2 × 10^4^	0.086
IV (*n* = 17)	12.58	2.89–4.0 × 10^5^	<*0.001*	14.58	3.05–6.1 × 10^5^	<*0.001*

^1^ Multivariable models regarding OS were adjusted for the tumor site, histological grade, and TNM stage. ^2^ Hazard ratio, estimated from proportional hazard Cox regression models. ^3^ Bias-corrected and accelerated 95% confidence interval of the estimated HR. ^4^ Statistically significant *p*-values are shown in italics.

**Table 5 ijms-21-08867-t005:** Characterization of the 168 CRC cases.

Variables	Number of Patients (%)
**Gender**	
Male	89 (53.0%)
Female	79 (47.0%)
**Tumor site**	
Colon	111 (66.1%)
Rectum	57 (33.9%)
**Tumor sidedness**	
Left	115 (68.5%)
Right	53 (31.5%)
**Histological grade**	
I	15 (8.9%)
II	129 (76.8%)
III	24 (14.3%)
**Venous invasion (137 of 168 patients)**	
Absent	118 (86.1%)
Present	19 (13.9%)
**Lymphatic invasion (137 of 168 patients)**	
Absent	121 (88.3%)
Present	16 (11.7%)
**T (tumor invasion)**	
T1	4 (2.4%)
T2	20 (11.9%)
T3	107 (63.7%)
T4	37 (22.0%)
**N (nodal status)**	
N0	95 (56.5%)
N1	42 (25.0%)
N2	31 (18.5%)
**TNM stage**	
I	20 (11.9%)
II	72 (42.9%)
III	59 (35.1%)
IV	17 (10.1%)

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
