# Peer review of "Identification of Two Novel Circular RNAs Deriving from BCL2L12 and Investigation of Their Potential Value as a Molecular Signature in Colorectal Cancer"

_ijms, 2020, doi:10.3390/ijms21228867_

Round 1

Reviewer 1 Report

Authors identified two novel circRNAs derived from BCL2L12 and investigated about their expression in CRC model and their potential as prognosis biomarkers.

The research paper is poor of experiments and there are no figuers relative about the experiments the authors  have performed. This make difficult undestand data and verify the data reported. The scientific language is not suitable and it is difficult to interpret. 

1) Results

  • 2.1 paragraph: the authors used an unique table (n 1) to report clinicopathological characteristics of patients of the study (this is not a result and it should be moved to matherial and methods) and to show expression levels of the two circRNAs (this should be show as a plot and in a separeted table).
  • 2.3 paragraph: the authors didn't show a table reported the name of the miRNAs predicted and probability scores. it is not clear which prediction tools they use and which tool gave good results.
  • Figure 1 is difficult to understand.
  • Table 2: the reader can not undestand the data reported in tha table. Make more explicative the legend. Moreover, show better data in the table, showing the matching of the results in the tools asked.
  • 2.4 paragraph: show more details about the results and the experiment performed. the reader can not understand which samples have been used (fresh frozen samples, cell lines...).
  • 2.5 paragraph: the reader can not understand the data because there is no a related figure. Moreover, in lane 149 it is impossible to undestand how many cell lines showed the results mentioned.
  • 2.6 paragraph: there is no a figure concerned to data reported.
  • 2.7 paragrapg: the authors must specify how many patients were included in the kaplan meier curves analysis.

2) Discussion

  • row 193: which database the authors used to predict the existance of the circRNAs studied?
  • row 216: how do the authors predicted miR-1915-5p? There is a lack of informations.
  • row 228: "This information indicates that circ_BCL2L12_1 affects CRC cells multifariously": This is only a prediction!
  • row 236-239: specify the reference in the text abput the information mentioned.
  • row 255: report how the prediction have been performed and show the results. Moreover, this in only a prediction, so the authors must not be sure that the circRNAs can be used as biomarkers. They need to performe a deeper analisys.

3) Matherials and methods

  • 4.2 paragraph: "One hundred and sixty-eight malignant colorectal tumors and 63 paired non-cancerous colonic tissue specimens": in row 101 the patients reported are 151. Is this a different cohort?
  • 4.3 paragraph: what kind of samples have been used? fresh frozen? FFPE? The authors did not give information about it.
  • 4.4 paragraph: in row 289 the authors should specify why they use a cell line pool and they should refere to a table or a figure about the divergent primers they mentioned.
  • row 303: make more informative table S2. The readers can not understand which primers have been used for.
  • 4.5 paragraph: there are no figures about the electroforesis gel and the sequencing. Specifiy which samples have been used for both experiments.
  • row 317: show miRNA prediction results in a table and specify the matching between different tools used.
  • row 323: there are no figures reported the results mentioned in the text.
  • row 325: "Due to the circular structure of these molecules, each circRNA sequence was submitted twice in all tools, starting from a different site each time" the authors explain what does it means.
  • row 335: there are no figures relative to real time experiments.
  • row 345: table S3-4 are confused and the reader can not understand data reported.
  • row 347: what kind oh test authors mean?

Author Response

Reviewer #1 (Comments to the Author):

  1. Results; 2.1 paragraph: the authors used a unique table (n 1) to report clinicopathological characteristics of patients of the study (this is not a result and it should be moved to material and methods) and to show expression levels of the two circRNAs (this should be shown as a plot and in a separated table).

As suggested by the Reviewer, we split that table into two:

  • Page 7, lines 157-159 (Results): Table 3 shows the distributions of the expression levels of the BCL2L12 circRNAs in cancerous and non-cancerous tissue samples. These data are also graphically illustrated in Figure S1.
  • Page 13, lines 288-289 (Materials and Methods): Table 5 describes the clinicopathological characteristics of CRC patients.

  1. Results; 2.3 paragraph: the authors didn't show a table reported the name of the miRNAs predicted and probability scores. it is not clear which prediction tools they use and which tool gave good results.

Regarding the Reviewer’s comment, we added two new Tables (Table 1 and Table S1):

  • Page 3, lines 112-113 (Results): Table 1: List of the miRNAs that are predicted to bind to the circRNAs using the miRDB tool, the prediction scores, and the motifs of the circRNAs that are predicted to sponge those miRNAs.
  • Table S1: miR-1915-5p–targeted mRNAs, the tools used to predict them, and the scores observed.

We also modified the text in Paragraph 2.2:

Page 4, lines 103-105 (Results): These miRNAs, the prediction scores, and the binding motifs of the circRNAs are shown in Table 1. The targets of miR-1915-5p, that were found in at least two of the four tools and have been reported to play a role in CRC, are shown in Table S1.

  1. Results; Figure 1 is difficult to understand.

Taking into consideration the Reviewer’s comment, we removed some data from Figure 1, in order to make it easier to understand; Figures 1a and 1b show the exon structures of the BCL2L12 circRNAs and the divergent primer pairs used in first-round PCR assay, while Figures 1c and 1d show the primer pairs used in semi-nested PCR assays. Figures 1e and 1f show the sequences of the circRNAs, starting from one of the exons between which the back-splice event takes place. Moreover, in Figure 1f, a predicted open reading frame is marked.

We also modified the respective Figure legend:

Page 4, lines 115-122 (Results): Figure 1. The structure and sequences of BCL2L12 circRNAs. Different colors indicate different exons; yellow: exon 1, light green: exon 2, light blue: exon 3, purple: exon 4, blue: exon 5, patterned blue: extended exon 5, dark green: exon 6. (a, b) Arrows outside the circles show the annealing positions of the primers used in the first-round PCR assay. (c, d) Arrows inside the circles indicate the annealing positions of the primers used in the semi-nested PCR assays, while arcs indicate the respective amplicons. (e, f) The sequences of BCL2L12 circRNAs, starting from the back-splice site. The colored fonts of the exon sequences match the colors used in the graphical illustrations of the circRNA exon structures. The black boxes show the putative translation start and stop codons of the predicted open reading frame (ORF). The ORF is underlined with a grey dash line.

  1. Results; Table 2: the reader cannot understand the data reported in the table. Make more explicative the legend. Moreover, show better data in the table, showing the matching of the results in the tools asked.

As suggested by the Reviewer, we explained which tool was used to retrieve the prediction data presented in Table 3:

Page 3, lines 106-109 (Results): In both circular RNAs, various protein-binding sites were detected and many RBPs were predicted to bind to them. RBPmap provided more information about the number of RBPs and their binding sites compared to beRBP; thus, we chose to use the results observed by this tool. We selected those RBPs with more than 5 binding sites, accompanied by a high probability score; these are presented in Table 2.

Moreover, we added more data in Table 3, concerning a score defining a significant match (Z-score), accompanied by a probability value (P value); these are explained in the footnotes. We also modified the respective legend:

Page 5, line 123 (Results): RNA binding proteins (RBPs) that are predicted to bind to each BCL2L12 circRNA.

  1. Results; 2.4 paragraph: show more details about the results and the experiment performed. the reader cannot understand which samples have been used (fresh frozen samples, cell lines...).

As suggested by the Reviewer, we provided more information about the experiments performed and the respective results:

Page 5, lines 128-133 (Results): A real-time qPCR assay was standardized for the specific BCL2L12 circRNA quantification. Aiming to assure the quantitative results of each assay, a standard curve was generated for each amplicon, using serial dilutions of polymerase chain reaction (PCR) products, deriving from a 25-cycle PCR assay. For circ-BCL2L12-1 and circ-BCL2L12-2 standard curves, PCR products from pre-amplification of Caco-2 cell line cDNA were serially diluted. For the HPRT1 standard curve, cDNA from a fresh frozen tissue specimen was pre-amplified and serially diluted.

  1. Results; 2.5 paragraph: the reader cannot understand the data because there is not a related figure. Moreover, in lane 149 it is impossible to understand how many cell lines showed the results mentioned.

We thank the Reviewer for this comment. In response to it, we added a new Figure (Figure 3, page 7, Results) showing the amplification plot of the real-time qPCR assay conducted using CRC cell lines as templates. Moreover, we modified the text appropriately:

Page 6, lines 145-149 (Results): Through the pre-amplification and qPCR assays, we relatively quantified the expression levels of both circRNAs in Caco-2, HCT 116, HT-29, COLO 205, SW 620, DLD-1, and RKO CRC cell lines. circ-BCL2L12-1 is present in all CRC cell lines, while circ-BCL2L12-2 was detected in 2 of them, namely Caco-2 and HCT 116. The expression levels of circ-BCL2L12-1 in these 2 cell lines were higher than those of circ-BCL2L12-2. The amplification plots of the qPCR assays for both circRNAs are shown in Figure 3.

  1. Results; 2.6 paragraph: there is not a figure concerned to data reported.

As suggested by the Reviewer, we made a graphical illustration of the data described in this paragraph. The data is presented in Figure S2.

  1. Results; 2.7 paragraph: the authors must specify how many patients were included in the Kaplan-Meier curves analysis.

We thank the Reviewer for this comment. We added the number of patients included in the Kaplan-Meier survival analysis:

Page 8, line 168 (Results): Survival data were available for 151 out of the 168 CRC patients.

  1. Discussion; row 193: which database the authors used to predict the existence of the circRNAs studied?

The circRNAs identified in this study were not recorded in any of the publicly available tools/databases. They were identified exclusively by the experimental procedure described in the manuscript. A respective phrase was added in the Discussion section, to clarify this issue:

Page 11, lines 194-196 (Discussion): however, none of these circRNAs have been experimentally validated yet, while the circRNAs identified in the current study have not been submitted to any of the available databases; thus no prediction data were available for the circRNAs identified in this study.

  1. Discussion; row 216: how do the authors predicted miR-1915-5p? There is a lack of information.

The binding of miR-1915-5p to circ-BCL2L12-1 was predicted by miRDB. A respective table (Table 1, page 3, lines 112-113, Results) was added showing this information, as well as a respective reference to this table.

Page 11, lines 219-220 (Discussion): As far as miRNAs are concerned, miR-1915-5p, which is predicted to be sponged by circ-BCL2L12-1 as shown in Table 1.

  1. Discussion; row 228: "This information indicates that circ_BCL2L12_1 affects CRC cells multifariously": This is only a prediction!

We agree with the Reviewer’s remark. Thus, we modified the text appropriately:

Page 11, lines 227-228 (Discussion): This information suggests a multifarious impact of circ-BCL2L12-1 on CRC cells; however, further investigation is required.

  1. Discussion; row 236-239: specify the reference in the text about the information mentioned.

This sentence was removed from the Discussion.

  1. Discussion; row 255: report how the prediction has been performed and show the results. Moreover, this is only a prediction, so the authors must not be sure that the circRNAs can be used as biomarkers. They need to perform a deeper analysis.

Following the Reviewer’s suggestion, we added the information about the prediction tools used and presented the results by adding new Tables (Table 1, in page 3; Table 2, in page 5).

Page 12, lines 249-250 (Discussion): Additionally, both circRNAs are predicted to bind miRNAs and RBPs, as revealed by miRDB and RBPmap tools, and shown in Table 1 and Table 2, respectively.

Regarding the potential biomarker utility, this conclusion was not reached using prediction tools, but by performing biostatistical analysis. This was also clarified in the revised manuscript:

Page 12, lines 254-256 (Discussion): The biostatistical analysis conducted revealed that circ-BCL2L12-1 could be used as a molecular biomarker of poor prognosis for the OS of CRC patients and can provide a better stratification for TNM II patients based on their OS intervals.

  1. Materials and methods; 4.2 paragraph: "One hundred and sixty-eight malignant colorectal tumors and 63 paired non-cancerous colonic tissue specimens": in row 101 the patients reported are 151. Is this a different cohort?

Following the Reviewer’s remark, we explained in the revised text that this is not a different cohort (i.e. validation cohort), but survival data were available for 151 out of the 168 patients:

Page 8, line 168 (Results): Survival data were available for 151 out of the 168 CRC patients.

Page 12, lines 275-277 (Materials and Methods): The patients were followed up for 52months (median time) and information regarding disease-free (DFS) and overall survival (OS) was collected, for 151 out of the 168 patients.

  1. Materials and methods; 4.3 paragraph: what kind of samples have been used? fresh frozen? FFPE? The authors did not give information about it.

We thank the Reviewer for this comment. We added the following information:

Page 12, lines 274-275 (Materials and Methods): One hundred and sixty-eight malignant colorectal tumors and 63 paired non-cancerous colonic fresh frozen tissue specimens were provided by University General Hospital “Attikon”.

  1. Materials and methods; 4.4 paragraph: in row 289 the authors should specify why they use a cell line pool and they should refer to a table or a figure about the divergent primers they mentioned.

Following the remark made by the Reviewer, the reason for using a cDNA pool was explained in paragraph 4.4. Moreover, the divergent primer pairs used are clearly depicted in Figures 1a and 1b (Page 4, Results) and in Tables S3 and S4.  Table S3 shows the primer sequences, while Table S4 shows the primer pairs used and the respective amplicon sizes.

Page 14, lines 291-292 (Materials and Methods): This pool was generated for experimental purposes, as this way, we were able to identify circRNAs expressed in any of the CRC cell lines.

  1. Materials and methods; row 303: make more informative table S2. The readers cannot understand which primers have been used for.

Following the Reviewer’s remark, we modified Tables S3 and S4, to render them more easily readable and understandable. (Please note that Table S2 of the original version is now Table S4).

Page 14, lines 304-306 (Materials and Methods): The sequences of the primers used for both assays are listed in Table S3, while the primer pairs and sizes of the amplicons observed are listed in Table S4.

  1. Materials and methods; 4.5 paragraph: there are no figures about the electrophoresis gel and the sequencing. Specify which samples have been used for both experiments.

Prompted by the Reviewer’s comment, we included the electropherograms of Sanger sequencing as Supplementary Information. We also clarified in the text which samples have been used for each experiment, to help the readers better understand our experimental workflow. Thus, we added the following sentences in the revised manuscript:

Page 14, lines 312-313 (Materials and Methods): Subsequently, the sequence was identified by Sanger sequencing (Supplementary Information).

Page 14, lines 308-309 (Materials and Methods): After the PCR assays, the PCR products observed through the amplification of the cDNA pool were loaded on a 2% agarose gel, stained with ethidium bromide.

  1. Materials and methods; row 317: show miRNA prediction results in a table and specify the matching between different tools used.

Following the Reviewer’s remark, a new Table (i.e. Table S1) including these data was added and appropriately mentioned in the revised manuscript:

Page 14, lines 318-319 (Materials and Methods): We then selected the targets that were identified in at least 2 of the tools and have been reported to play a role in CRC, as shown in Table S1.

  1. Materials and methods; row 323: there are no figures reported the results mentioned in the text.

Prompted by the Reviewer’s comment, the predicted open reading frame (ORF) was underlined in Figure 1f (Page 4, Results), with a grey dash line; the putative start and stop codons are also now marked with black boxes. These annotations are explained in detail in the Figure 1f legend.

  1. Materials and methods; row 325: "Due to the circular structure of these molecules, each circRNA sequence was submitted twice in all tools, starting from a different site each time" the authors explain what it means.

We thank the Reviewer for this remark. We added the following text to explain what it means:

Page 14, lines 326-330 (Materials and Methods): In order to carry out these predictions, the sequence of the circRNA should be submitted as linear. Therefore, we should “cut” the sequence in a stochastic position. Aiming to assure that no information for binding sites will be lost, due to this decision, we conducted all of the predictions twice, following the cleavage of circRNA sequence at a different point. More specifically, the last exon of the sequence which was submitted the first time in each prediction was transferred at the beginning of the sequence for the following predictions.

  1. Materials and methods; row 335: there are no figures relative to real time experiments.

An amplification plot from the real-time qPCR assay was added (Figure 3, page 7, Results). Moreover, standard curves that were built during the standardization of the real-time qPCR assays and melt curves of the respective amplicons are now depicted in Figure 2 (page 6, Results). We also added the respective references in the revised manuscript:

Page 15, lines 338-341 (Materials and Methods): A real-time qPCR assay was developed using the KAPA SYBR FAST qPCR Master Mix (2X) Kit (KAPA Biosystems Inc., Woburn, MA, USA), as previously described (Figure 3). A standard curve was generated for each amplicon, using serial dilutions of PCR products. A graph was built by plotting the threshold cycle (CT) versus the quantity (Figures 2a and 2b).

  1. Materials and methods; row 345: table S3-4 are confused and the reader cannot understand data reported.

Following the Reviewer’s remark, we modified Tables S5 and S6, to render them more easily readable and understandable. (Please note that Table S3 of the original version is now Table S5, and that Table S4 of the original version is now Table S6). 

Page 15, lines 345-346 (Materials and Methods): The sequences of the primers used are listed in Table S5, while the primer pairs and sizes of the amplicons observed are listed in Table S6.

  1. Materials and methods; row 347: what kind of test authors mean?

Prompted by the Reviewer’s comment, we added the following information to explain what we exactly mean:

Page 15, lines 348-350 (Materials and Methods): Due to the fact that the distribution in our cohort was not normal, non-parametric tests, such as the Wilcoxon signed-rank and Mann Witney U tests, were used. Both of them do not take into consideration the distribution in the cohort.

The authors wish to thank the Reviewers for their constructive comments that led to the improvement of the current manuscript.

Reviewer 2 Report

This is a large comprehensive study to identify BCL2L12-specific circRNAs in colorectal cancer. For that reason 6 human cancer cells lines and 168 CRC samples have been used. The authors found 2 circRNA _1 and_2 with some genetic similarities and differences, namely that _2 may have more protein binding partners...interestingly the two novel circRNAs have aboundant splicing partners...

The main observations are that circRNA_1 is decreased while circRNA_2 is upregulated in CRC tumors. However, only circRNA_1 has prognostic impact: its decrease define favourable prognosis. 

To be corrected:

MM: what was the source of RNA in case of tumor samples? FFPE or fresh frozen? What is the mutation status of the cancer cell lines used? (RAS/RAF/MSI/p53)?

Fig.3 is missleading by characterizing CRCs on circRNA_1 negative/positive, since the median expression level was used as cut-off, so low (decreased) and maintained/high is the correct definition!

Table 3. Multivariate analysis must be completed with a new independent prognostic factor of CRC which is sidedness (L/R). 

Missing analysis. CRC is molecularly a heterogenous group of tumors as it is stated. it has to be analysed if any of the novel circRNAs have any association with clinically relevant molecular subgroups: RAS or BRAF mutants, MMRdefficiency and may be p53 mutation....

Author Response

Reviewer #2 (Comments to the Author):

  1. MM: what was the source of RNA in case of tumor samples? FFPE or fresh frozen? What is the mutation status of the cancer cell lines used? (RAS/RAF/MSI/p53)?

We thank the Reviewer for this comment. We added the following information:

Page 12, lines 274-275 (Materials and Methods): One hundred and sixty-eight malignant colorectal tumors and 63 paired non-cancerous colonic fresh frozen tissue specimens were provided by University General Hospital “Attikon”.

Moreover, the mutational status of the CRC cell lines used in the current research study is presented in Table S2.

Page 12, line 269 (Materials and Methods): The mutational status of the CRC cell lines is shown in Table S2.

  1. 3 is misleading by characterizing CRCs on circRNA_1 negative/positive since the median expression level was used as cut-off, so low (decreased) and maintained/high is the correct definition!

As suggested by the Reviewer, we replaced the term “circ-BCL2L12-1-positive” with the term “circ-BCL2L12-1-high”, and the term “circ-BCL2L12-1-negative” with the term “circ-BCL2L12-1-low” in Figure 3 and throughout the text.

  1. Table 3. Multivariate analysis must be completed with a new independent prognostic factor of CRC which is sidedness (L/R).

In response to the Reviewer’s remark, we added the tumor sidedness (right vs. left) to the clinicopathological characteristics of the patients (Table 5, page 13, lines 288-289, Materials and Methods) and to the univariate Cox regression analysis (Table 4, page 10, lines 181-185, Results). However, as this factor did not have prognostic significance in the CRC patients’ cohort of our study, we could not include it in the multivariate Cox regression analysis (since adjustment of the multivariate analysis for this cofactor would lead to violation of a multivariate Cox regression assumption).

  1. Missing analysis. CRC is molecularly a heterogeneous group of tumors as it is stated. It has to be analyzed if any of the novel circRNAs have any association with clinically relevant molecular subgroups: RAS or BRAF mutants, MMR deficiency and may be p53 mutation...

We agree with the Reviewer that analyzing the associations of BCL2L12 circRNAs with other molecular characteristics of the CRC patients would be very interesting. Nevertheless, such data are not available for our cohort, unfortunately. Therefore, we decided to simply comment on this interesting aspect at the end of the Discussion section:

Page 12, lines 256-258 (Discussion): Moreover, it would be interesting to assess putative associations of these novel circRNAs and other molecular characteristics of the patients, for instance, RAS or BRAF or TP53 mutations, and MMR deficiency.

The authors wish to thank the Reviewers for their constructive comments that led to the improvement of the current manuscript.

Round 2

Reviewer 1 Report

The authors improve the study by adding all the suggestions given by the reviewer.